# Effect of Operational Variables on the Yield of Chemoenzymatic Oxidation of 2,5-Furandicarboxaldehyde to 2,5-Furandicarboxylic Acid in Fed-Batch and Continuous Packed-Bed Millibioreactor

Cristian Balboa [1], Rodrigo A. Schrebler [2], María Elena Lienqueo [3] and Nadia Guajardo [4,*]

[1] Escuela de Química, Universidad Tecnológica Metropolitana, Las Palmeras 3360, Ñuñoa, Santiago 7800003, Chile
[2] IONCHEM SpA, El Tordillo, 154, Villa Alemana, Valparaíso 2471548, Chile
[3] Centro de Biotecnología y Bioingeniería (CeBiB), Departamento de Ingeniería Química, Biotecnología y Materiales, Universidad de Chile, Beauchef 851, Santiago 8370456, Chile
[4] Programa Institucional de Fomento a la Investigación, Desarrollo e Innovación, Universidad Tecnológica Metropolitana, Ignacio Valdivieso 2409, San Joaquín, Santiago 8940000, Chile
* Correspondence: nguajardo@utem.cl

**Abstract:** This work explores for the first time the use of a fed-batch and continuous packed-bed millibioreactor for the chemoenzymatic oxidation of 2,5-furandicarboxaldehyde (DFF) to 2,5-furandicarboxylic acid (FDCA). Different operational variables were studied: temperature, substrate concentration, and flow rate using different reactors (batch, fed-batch, and a continuous packed-bed bioreactor). The best yield (100%) was achieved using the fed-batch reactor at an $H_2O_2$ flow rate of 3 μL/min with a substrate concentration (DFF) of 100 mM. Regarding the specific productivity, the highest values (>0.05 mg product/min g biocatalyst) were reached with the operation in the fed-batch bioreactor and the continuous packed-bed bioreactor. The yield of the biocatalyst decreased by 98% after the first reaction cycle during the operational stability tests, due to a substantial inactivation of the biocatalyst by $H_2O_2$ and peracid. In this study, it is possible to select the operational variables in fed-batch and continuous reactors for chemoenzymatic oxidation that can increase the yield and specific productivity; however, the stability of the biocatalyst should be improved in future research.

**Keywords:** 2,5-furandicarboxaldehyde; 2,5-furandicarboxylic acid; chemoenzymatic oxidation; continuous oxidation; fed-batch oxidation

## 1. Introduction

Due to the depletion of non-renewable natural resources such as petroleum-based oil, biorefineries are expected to be central in the manufacturing of biogenic raw materials, such as chemical products and fuels. Numerous investigations in the academic and industrial fields give an account of the efforts to replace non-renewable sources with renewable ones [1–4].

Lignocellulosic residues are an essential raw biomass source for obtaining highly functionalized building blocks using different valorization strategies [1–3]. These compounds of great interest are furfural (FF) and 5-hydroxymethylfurfural (HMF), which correspond to byproducts of lignocellulosic pre-treatments [5,6]. The conventional way of synthesizing FF and HMF is by triple acid-dehydration of pentoses to obtain FF and hexoses to produce HMF [7,8]. The interest is that these molecules can be functionalized in different ways to obtain building blocks for creating new products [9–13].

Complete oxidation of HMF results in a 2,5-furandicarboxylic acid (FDCA). FDCA is a compound of lavish attention because through its polymerization with ethane 1,2 diol, it forms polyethylene furoate (PEF), which may replace the petroleum-based PET [14–16] and is considered within the top twelve building blocks. In addition, a promising application

of FDCA has been reported as a biobased acid catalyst for the biphasic fractionation of lignocellulose [17].

The complete oxidation of HMF to FDCA is not trivial; it involves three consecutive oxidation steps (Figure 1a). Because the classical synthesis of FDCA is expensive for using metals as catalysts [18,19] and environmentally unfriendly due to the use of high pressure, high temperatures, and volatile and hazardous organic solvents [19–23], new techniques have been developed as inexpensive inorganic bases [24], lithium hydroxide [25], oxygen [26], enzymes [14,27–31], and whole cells [32,33].

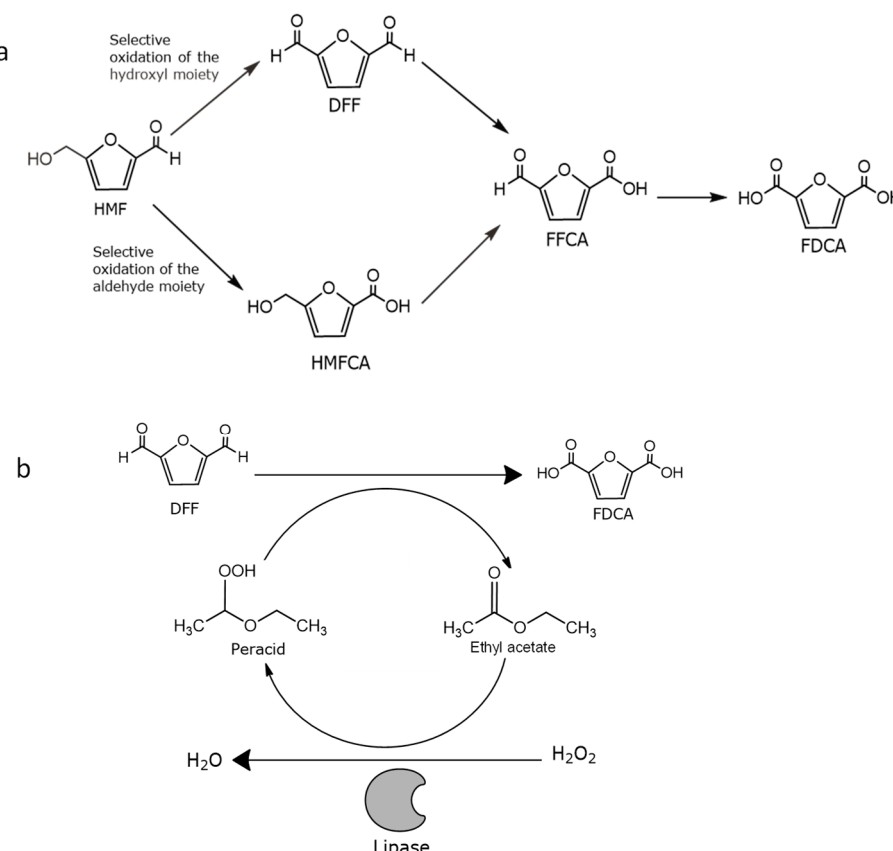

**Figure 1.** The pathway to produce FDCA. (**a**) From HMF and (**b**) chemoenzymatic oxidation of 2,5-furandicarboxaldehyde (DFF) to 2,5-furandicarboxylic acid (FDCA). HMFCA: 5-hydroxymethyl-2- furan carboxylic acid, FFCA: 5-formyl-2- furancarboxylic acid.

Direct bio-oxidation of HMF to FDCA is challenging to carry out. For this reason, and because the biotransformation of HMF to DFF is selective and is well documented [30], an exciting alternative is to oxidize chemoenzymatically the DFF to FDCA [29,34] (Figure 1b). This type of oxidation has already been reported in batch mode with yields close to 100% [29,35]. However, to our knowledge, fed-batch and continuous mode operation has not been yet carried out. The control of the operational variables and the operation of reactors in different modalities can increase the yield of the reaction, reducing the inactivation by substrate ($H_2O_2$) and product (peracid) of the biocatalyst [36,37]. Because the enzymatic reactions carried out in fed-batch and continuous bioreactors reach higher specific productivities [38–42], the main objective of this work is to study for the first time the effect of operational variables, such as flow, temperature, and concentration of substrates (DFF and $H_2O_2$) on the yield of the chemoenzymatic oxidation of DFF to FDCA in a fed-batch and continuous packed-bed millibioreactor.

## 2. Material and Methods

### 2.1. Chemicals

2,5-furandicarboxaldehyde (DFF) and 2,5-furandicarboxylic acid (FDCA) were purchased from Sigma-Aldrich and were used without modification. Ethyl acetate, HPLC-grade methanol, formic acid, and $H_2O_2$ (30%) were purchased by Merck, Chile S.A. The immobilized lipase B from *Candida antarctica* (Novozym 435®) was kindly donated by Blumos S.A. (Chile). The enzymatic activities reported by the manufacturers in the tributyrin hydrolysis reaction are 2000 U/g.

### 2.2. Analytical Method for the Quantification of FDCA

The reaction products were quantified by HPLC (JASCO LC- 4000 with a diode array detector) with a GL Sciences $C_{18}$ HPLC column (100 × 4.6 mm I.D) at 264 nm under the following conditions: mobile phase was composed of solutions A and B at a volume ratio A/B = 85/15, where A = water/formic acid (99.5/0.5) and B = methanol (100). The flow rate was 0.6 mL/min at 25 °C and the retention time for DFF was 4.5 min and FDCA 7.5 min.

### 2.3. Chemoenzymatic Synthesis of FDCA in a Batch Bioreactor

The chemoenzymatic oxidation of DFF to furnish FDCA in batch operation was carried out in a sealed reactor (20 mL) in a shaker at 165 rpm and 40 °C. The reaction media was prepared by mixing DFF (10 mM) with 0.2 mL of 30% $H_2O_2$ in 9.8 mL of ethyl acetate for 48 h. The reaction conditions to be evaluated are detailed in Table 1.

**Table 1.** Reaction conditions are to be evaluated.

| Reaction Conditions | Range |
|---|---|
| Concentration of DFF (mM) | 10–50 mM |
| Biocatalyst loading (mg)/Activity (U) | 100–835 mg/20–167 U |
| Amount of $H_2O_2$ 30% (μL) | 120–1021 μL |

The reaction started by adding 0.167 g of Novozym 435®, which has already been used in perhydrolysis reactions [37]. Reaction samples were taken at different times and they were diluted with a mobile phase before analysing the reaction products on HPLC. The experimental tests were carried out in duplicate. The yield was calculated with the following Equation (1):

$$Yield\ (\%) = \left[ \frac{n_p - n_{po}}{n_{so}} \cdot \frac{v_s}{v_p} \right] * 100 \tag{1}$$

where $p$ = product; $n_{P_0}$ = amount of product P at the start of the reaction (mol); $n_{S_0}$ = amount of substrate S at the start of the reaction (mol); $n_P$ = amount of product P at the end of the reaction (mol); $v_S$ = stoichiometric factor for substrate S; and $v_P$ = stoichiometric factor for product P.

### 2.4. Chemoenzymatic Synthesis of FDCA in Fed-Batch Bioreactor

The chemoenzymatic oxidation of DFF to FDCA was carried out in a sealed reactor (20 mL) using 1 g of Novozym® 435. The $H_2O_2$ (2043 μL) was fed using a syringe pump at a flow of 1 to 5 μL/min. The reaction volume was 10 mL, and the reaction media were prepared by mixing 10 mL of ethyl acetate and DFF (100 mM). The reaction was carried out in a rotary incubator at 160 rpm and 40 °C (to prevent the decomposition of $H_2O_2$). Reaction samples were taken at different times to analyze the reaction products. The conversion was calculated according to Equation (1). The experimental tests were carried out in duplicate.

### 2.5. Chemoenzymatic Synthesis of FDCA in a Continuous Packed-Bed Millibioreactor

The chemoenzymatic synthesis of FDCA in a continuous packed-bed reactor was carried out in a column (0.4 cm of diameter × 12.6 cm of length) filled with 0.5 g of

biocatalyst, as shown the Figure S1. The reaction media was formed by 50 mM of DFF and 1021 μL of $H_2O_2$. The reaction operation was carried out at a flow of 7–20 μL/min at 30 °C. The samples were withdrawn/extracted at different times to analyze the reaction products and the experimental tests were carried out in duplicate. The residence time ($\tau$) was determined according to Equation (2).

$$\tau = \frac{\epsilon \, V_t}{q} \tag{2}$$

where $\epsilon$ denotes the void fraction, $V_t$ is the total bed volume, and q is the substrate flow rate. The void fraction of the packed-bed reactor was calculated with Equation (S1), resulting in c.a 0.42 [43].

*2.6. Operational Stability of the Biocatalyst*

The stability of the biocatalyst under operating conditions was carried out in fed-batch mode. In fed-batch mode, the reaction was started by adding 2043 μL of $H_2O_2$ using a syringe pump at a flow of 1.46 μL/min. After 24 h, the biocatalyst was recovered by filtration. The biocatalyst was washed three times with a 100 mM phosphate buffer of pH seven and samples of the reaction medium were taken to determine the concentration of FDCA by HPLC (details in Section 2.2 "Analytic method for the quantification of FDCA"). The yield was calculated using Equation (1).

**3. Results and Discussion**

*3.1. Chemoenzymatic Synthesis of FDCA in a Batch Reactor*

Figure 2 details the chemoenzymatic oxidations in a batch reactor at different conditions. The mg of biocatalyst and volume of $H_2O_2$ in each graph of Figure 2 are different for the purpose of evaluating different conditions. For example, Figure 2A shows the results of the effect of the biocatalyst load and the amount of $H_2O_2$ over the yield at a concentration of 10 mM of DFF, reaching the highest yield (80%) with 167 mg (33 U) of biocatalyst and 200 μL of $H_2O_2$. Here, it can be seen that an increase in the amount of $H_2O_2$ decreases the yield of the reaction due to the inactivation of the biocatalysis. The peracid also causes the inactivation of the biocatalyst in the reaction media formed by the perhydrolysis reaction of ethyl acetate with $H_2O_2$ catalyzed by the lipase immobilized. This behavior has already been reported in several investigations related to the inactivation of the biocatalyst due to the high concentrations of $H_2O_2$ and peracid [34,36,37]. The researchers point out that these drawbacks can be improved by selecting the reaction's operational conditions [36], such as the modification of the biocatalyst by immobilization and protein engineering [34].

Figure 2B details the study's results of the effect of biocatalyst loading on reaction yield at a substrate concentration of 30 mM. An increase in the amount of biocatalyst reduces the yield of the reaction, probably due to the rise in peracid that causes the inactivation of the biocatalyst. Figure 2C details the results of the study of the effect of biocatalyst loading on reaction yield at a substrate concentration of 50 mM. In this case, the behavior of the reaction kinetics is similar to that in Figure 2B, since an increase in the amount of biocatalyst decreases the reaction yield. Comparing the three graphs in Figure 2, the highest yield is reached at a substrate concentration of 10 mM.

We evaluated different operational variables for fed-batch and continuous bioreactors in this work. According to previous research, these variables were selected because they were the most relevant [36,37]. The results are detailed in the following sections.

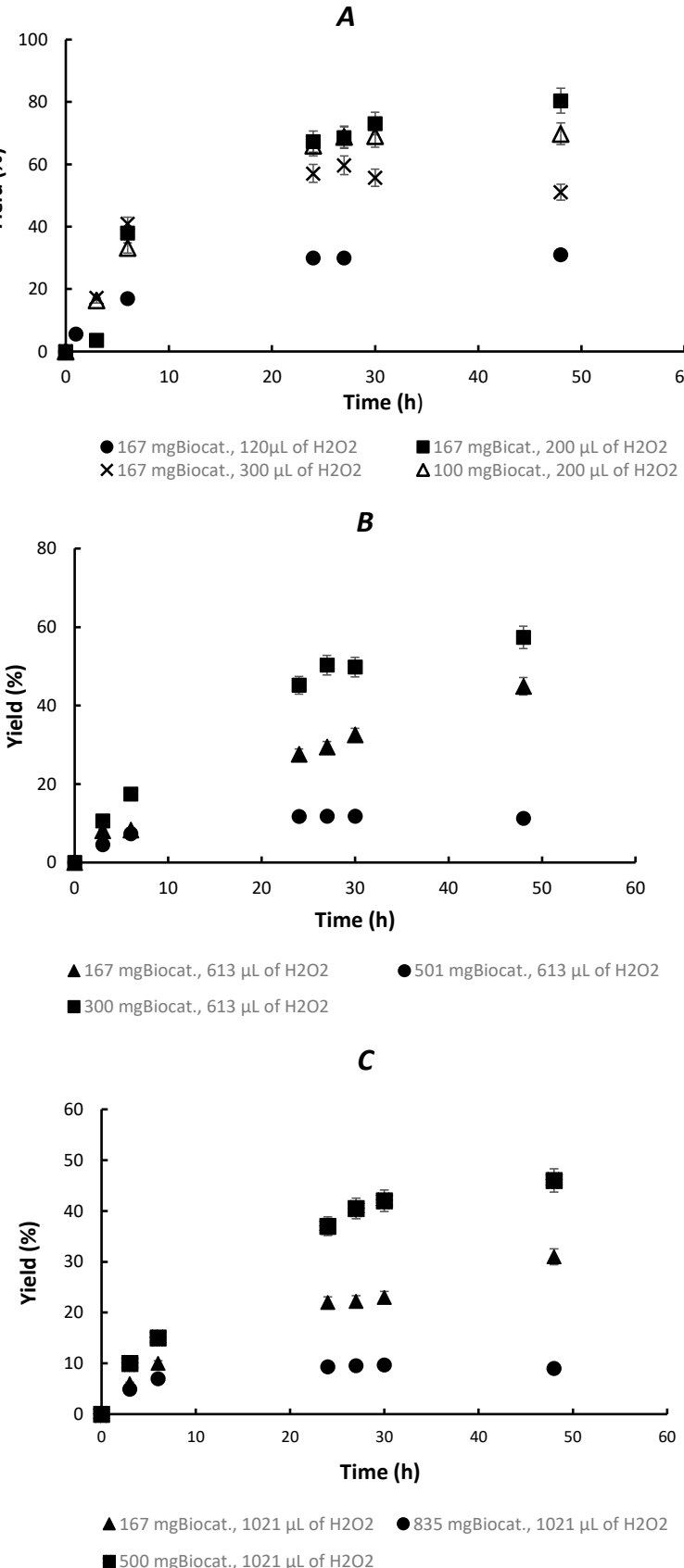

**Figure 2.** Chemoenzymatic synthesis of FDCA in a batch reactor. Operational conditions: 10 mL of reaction volume and 40 °C. (**A**) 10 mM of DFF; (**B**) 30 mM of DFF; (**C**) 50 mM of DFF.

### 3.2. Chemoenzymatic Synthesis of FDCA in Fed-Batch Bioreactor

One of the alternatives to increase the yield of the reaction is the operation of a fed-batch reactor. In this mode of operation, hydrogen peroxide is fed gradually with the help of a pump to reduce the inactivation of the biocatalyst, unlike the batch bioreactor, in which the entire volume of $H_2O_2$ is added to the reaction media, inactivating the biocatalyst faster.

Figure 3 shows the behavior of the chemoenzymatic oxidation in stepwise mode. Under the conditions of 50 mM DFF, 500 mg biocatalyst, and 1021 μL $H_2O_2$ added at the beginning, after 6 h and after 24 h, the yield of the reaction increased by 20% compared to the operation in batch mode. These results are due to the reduction in the inactivation of the biocatalyst, due to the gradual addition of the oxidant.

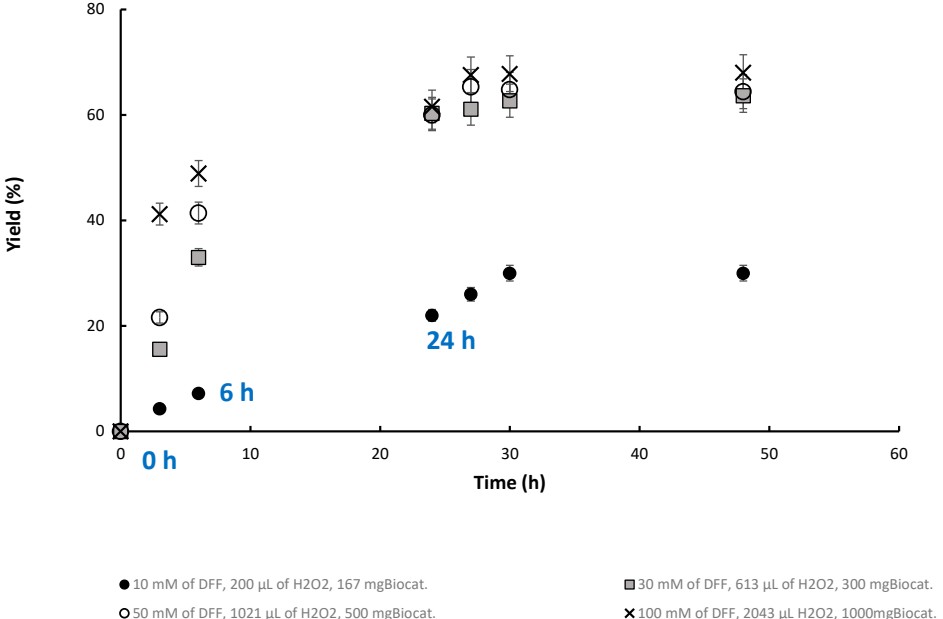

● 10 mM of DFF, 200 μL of H2O2, 167 mgBiocat.   ■ 30 mM of DFF, 613 μL of H2O2, 300 mgBiocat.
○ 50 mM of DFF, 1021 μL of H2O2, 500 mgBiocat.   ✕ 100 mM of DFF, 2043 μL H2O2, 1000mgBiocat.

**Figure 3.** Chemoenzymatic synthesis of FDCA with stepwise addition of $H_2O_2$. Operational conditions: 10 mL of reaction volume and 40 °C. The $H_2O_2$ addition was divided into three portions along the reaction: at the beginning, 6 h, and at 24 h.

To further increase the yield of the reaction, experimental tests were carried out in a fed-batch reactor using different flows of $H_2O_2$. Figure 4A details the results of the effect of the DFF concentration in a fed-batch bioreactor at an $H_2O_2$ flow rate of 1 μL/min. This graph shows a slight difference in the yield of the reaction kinetics carried out at 50 mM and 100 mM. Figure 4B shows the results of the effect of the DFF concentration on the reaction yield at a flow rate of 3 μL/min. As in Figure 4A, the behavior of the reaction kinetics is similar for both concentrations. However, at the end of 50 h, the reaction at 100 mM of DFF reaches a higher yield (100%) than that carried out at 50 mM of DFF. Using a flow of 5 μL/min (Figure 4C), there is no marked effect of the substrate concentrations (DFF) on the yield, which reached almost 100% at 30 h of reaction. Using a flow rate of 1 and 5 μL/min, the yield was 20% lower than at 3 μL/min. This could be because at 1 μL/min, there is a longer residence time of $H_2O_2$ and peracid in the catalytic bed, inactivating the biocatalyst. In comparison, at 5 μL/min, the residence time of the substrates in the catalytic bed is minor, decreasing the reaction yield.

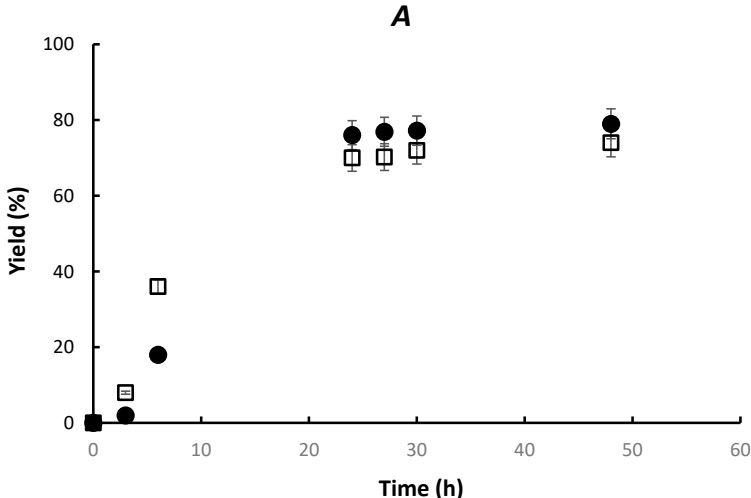

● 50 mM of DFF, 1021 µL of H2O2, 500 mg Biocat.   ☐ 100 mM of DFF, 2043 µL of H2O2, 1000 mg Biocat.

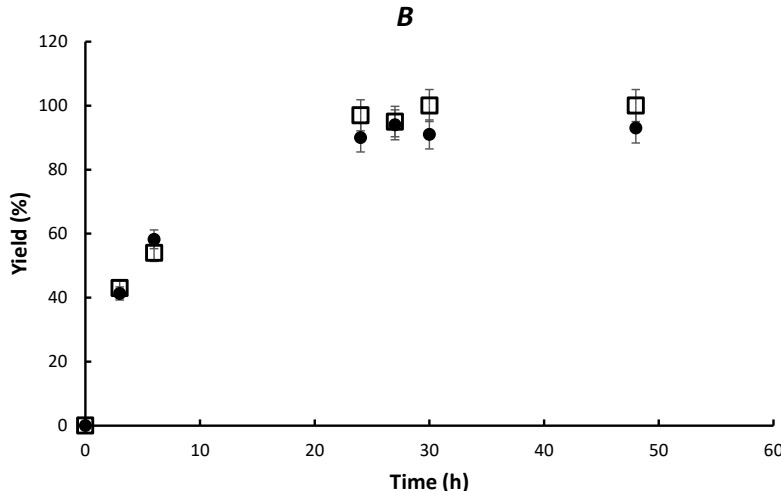

● 50 mM of DFF, 1021 µL of H2O2, 500 mgBiocat.   ☐ 100 mM of DFF, 2043 µL of H2O2, 1000 mgBiocat.

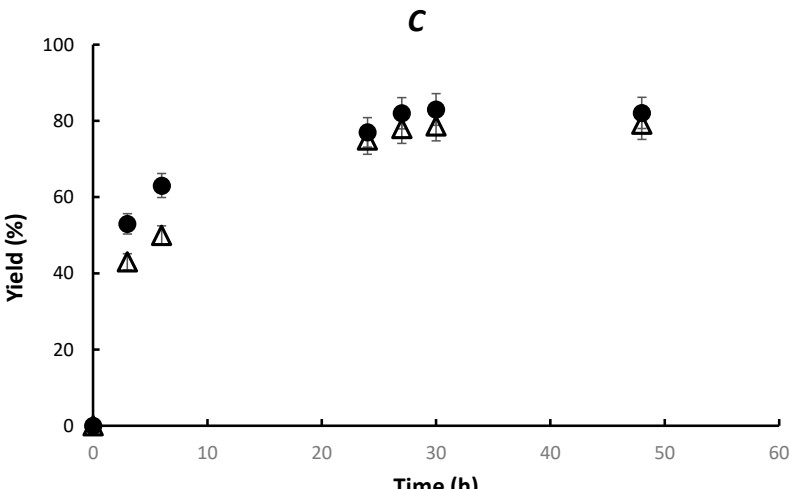

● 50 mM of DFF, 1021 µL of H2O2, 500 mgBiocat.   △ 100 mM of DFF, 2043 µL of H2O2, 1000mgBiocat.

**Figure 4.** Chemoenzymatic synthesis of FDCA in a fed-batch bioreactor. Operational conditions: 10 mL of reaction volume and 40 °C. One inlet at a flow rate of $H_2O_2$: (**A**) 1 µL/min; (**B**) 3 µL/min; and (**C**) 5 µL/min.

### 3.3. Chemoenzymatic Synthesis of FDCA in a Continuous Packed-Bed (CPBR) Bioreactor

Figure 5 shows the results of the chemoenzymatic synthesis of FDCA in a continuous packed-bed bioreactor *(CPBR)*. Figure 5A shows the effect of flow rate and the amount of $H_2O_2$ on the yield in a continuous packed-bed bioreactor at 40 °C. In this case, the condition that maximizes the yield is using 200 μL of $H_2O_2$ and a flow rate of 20 μL/min. Figure 5B shows the effect of the flow rate and the amount of $H_2O_2$ on the yield at 30 °C. With the use of an intermediated flow rate of 10 μL/min and 511 μL of $H_2O_2$, it is possible to achieve a stable yield behavior, approaching the steady state in the bioreactor.

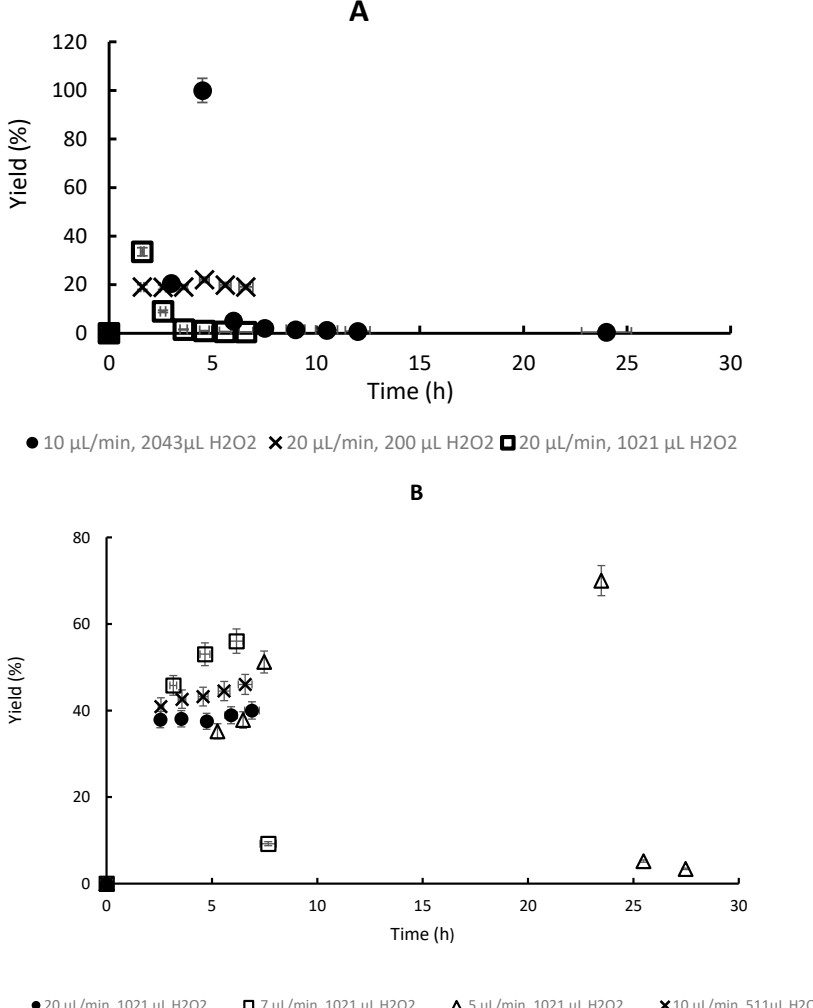

**Figure 5.** Chemoenzymatic synthesis of FDCA in a continuous packed-bed bioreactor catalyzed by Novozym 435 at a different flow rate. (**A**) 40 °C; (**B**) 30 °C. Reactions conditions: 50 mM of DFF, 500 mg of biocatalyst, and 10 mL of reaction volume except (**x**) reaction with 5 mL of reaction volume.

Under this mode of operation, the inactivation of the biocatalyst becomes evident, which causes the behavior to deviate from the ideal; it is not reaching a steady state. For example, at 40 °C, 10 μL/min, and 2043 μL of $H_2O_2$, a maximum yield of 100% is reached after 5 h; then, it decreases to 0% after 24 h. The best result (20%) was achieved at a flow rate of 20 μL/min and 200 μL of $H_2O_2$ for approximately 7 h of reaction.

Due to the synergistic effect of temperature with hydrogen peroxide and peracid [36,37], the reaction was carried out at 30 °C at 10 μL/min for 7 h (Figure 5B).

As can be seen in Figure 5, the reaction in the continuous packed-bed bioreactor at 30 °C increases the yield by 20% at 10 μL/min and 511 μL of $H_2O_2$. In similar reactions at 40 °C, yields remain stable for almost 7 h.

The ideal operation in a continuous packed-bed bioreactor is that the steady state is reached within three or six residence times. One of the conditions by which the steady state is not achieved is the inactivation of the biocatalyst. For example, Figure S2 shows that the residence times are longer below 20 $\mu$L/min because the biocatalyst is more exposed to inactivation by the substrate and products.

It was also tested by feeding the biocatalytic column's two inlets (Figure S1B): one for the ethyl acetate and DFF feed at 10 $\mu$L/min; and the other for the hydrogen peroxide feed at 1 $\mu$L/min. Figure 6 indicates a deviation from ideality during the first 5 h to reach approximately 50% of the yield at 7 h. The yield at 24 h was also verified, which decreased by 98% (data not shown).

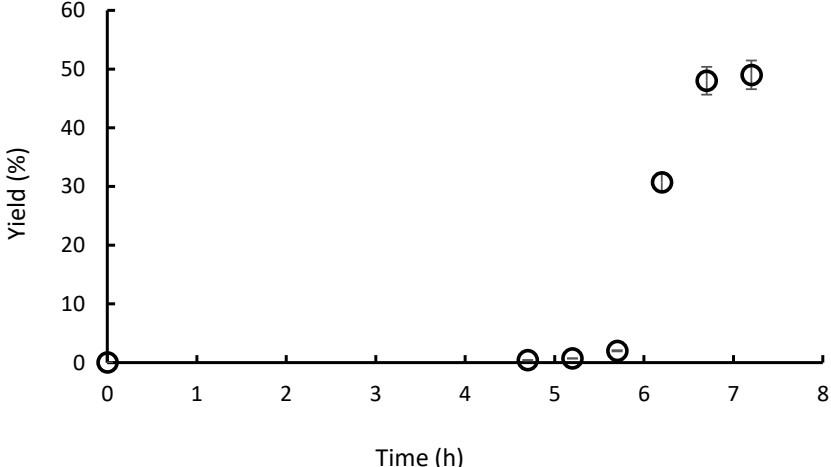

**Figure 6.** Chemoenzymatic synthesis of FDCA in a continuous packed-bed bioreactor catalyzed by Novozym 435 with two feeds. Reaction conditions: 30 °C, 5 mL of ethyl acetate at 50 mM of DFF at 10 $\mu$L/min, and 511$\mu$L of $H_2O_2$ at 1 $\mu$L/min.

According to the results, the chemoenzymatic oxidation in continuous mode should be carried out for 8 h to approach the steady state.

Specific productivities for the chemoenzymatic reactions using 50 mM of DFF were compared (Figure 7); the continuous and semi-continuous reactors reached twice the productivity of the batch reactor. This behavior may be due to reduced biocatalyst inactivation by the presence of $H_2O_2$ and peracid.

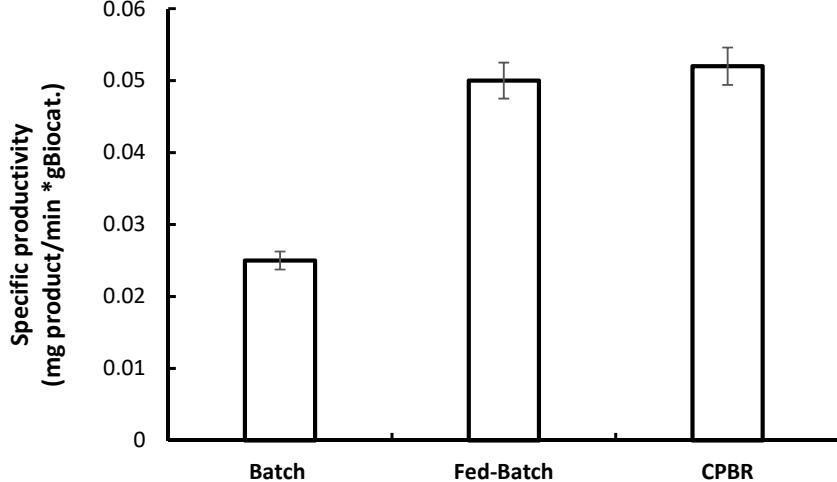

**Figure 7.** Comparison of productivities of chemoenzymatic oxidation to produce FDCA in different reactors, using a substrate concentration of 50 mM of DFF, 40 °C, 500 mg of biocatalyst, and 1021 $\mu$L of $H_2O_2$.

### 3.4. Operationalss Stability of the Biocatalyst in a Fed-Batch Bioreactor

Operational stability is essential for the operation of robust continuous reactors and for reducing the costs of the process associated with the biocatalyst. The operational stability of the biocatalyst in the fed-batch bioreactor was determined because it is comparable with the continuous packed-bed bioreactor; that is, if the biocatalyst is inactivated in a fed-batch bioreactor, the biocatalyst will also be inactivated in a continuous packed-bed bioreactor. Figure 8 details the reuse of the biocatalyst. As can be seen, the biocatalyst activity decreases sharply (98%) after the first cycle, due to the presence of $H_2O_2$ and peracid for both the bio-oxidation at 50 mM and 100 mM of DFF.

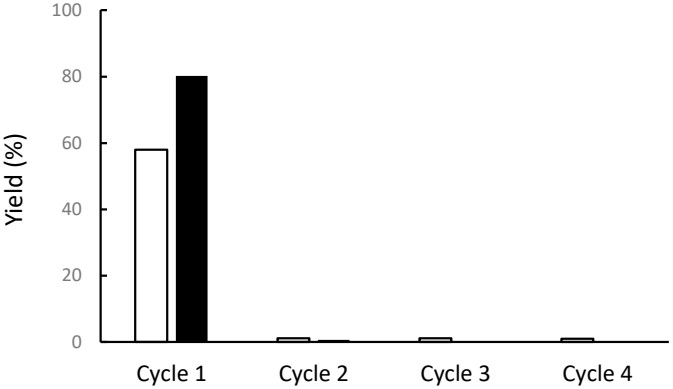

☐ 40°C, 100 mM, 1 gBiocat. and 3 µL/min   ■ 30°C , 50 mM DFF, 0.5 gBiocat. and 3 µL/min

**Figure 8.** Operational stability of the biocatalyst in the chemoenzymatic oxidation for the synthesis of FDCA in fed-batch mode. Operating conditions: 10 mL of the reaction medium, 2043 µL of $H_2O_2$ at a flow of 1.46 µL/min, and 24 h of reaction.

The activity decay is more pronounced in the reaction carried out at 50 mM DFF at an $H_2O_2$ flow of 3 µL/min, possibly because the residence time of the substrates and products inside the column are more significant, generating the inactivation of the biocatalyst. According to these results, the biocatalyst in batch and fed-batch modes can only be used in one reaction cycle and in continuous mode for approximately 8 h of reaction.

### 4. Conclusions

The operation of the chemoenzymatic oxidation of DFF to produce FDCA in a fed-batch and continuous packed-bed millireactor was successfully demonstrated for the first time. The fed-batch process led to yields of 54% higher compared to the batch bioreactor. In terms of productivity, the fed-batch and packed-bed bioreactor achieved similar results. The biocatalyst stability in the fed-batch mode decreased by approximately 98% after the first cycle. In contrast, in the continuous packed-bed millireactor, the biocatalyst stability remained stable during 8 h of reaction at a flow of 10 µL/min. This work demonstrates that continuous and fed-batch reactors can increase yield and productivity in the chemoenzymatic synthesis of FDCA.

**Supplementary Materials:** The following supporting information can be downloaded at: https://www.mdpi.com/article/10.3390/pr10102095/s1, Figure S1: continuous packed-bed reactor to produce FDCA; Figure S2: Residence times in operation with continuously packed bed bioreactor.

**Author Contributions:** Conceptualization, N.G.; Data curation, N.G.; Formal analysis, R.A.S. and M.E.L.; Funding acquisition, N.G.; Investigation, C.B. and M.E.L.; Methodology, C.B. and R.A.S.; Software, C.B. and R.A.S.; Supervision, N.G.; Validation, C.B. and M.E.L.; Writing—original draft, N.G.; Writing—review & editing, M.E.L. All authors have read and agreed to the published version of the manuscript.

**Funding:** This research was funded by FONDECYT grant number 1200558.

**Institutional Review Board Statement:** Not applicable.

**Informed Consent Statement:** Not applicable.

**Data Availability Statement:** Not applicable.

**Acknowledgments:** N.G. acknowledges the financial support from Fondecyt N° 1200558 (Chile).

**Conflicts of Interest:** On behalf of all the authors, the corresponding author states that there is no conflict of interest.

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
