# Peer review of "Effect of Operational Variables on the Yield of Chemoenzymatic Oxidation of 2,5-Furandicarboxaldehyde to 2,5-Furandicarboxylic Acid in Fed-Batch and Continuous Packed-Bed Millibioreactor"

_processes, doi:10.3390/pr10102095_

Round 1
Reviewer 1 Report
Guajardo and co-workers present an innovative biocatalytic approach, based on semicontinuous and continuous packed bed millibioreactor, for the oxidation of 2,5-furandicarboxaldehyde (DFF) to 2,5-furandicarboxylic acid. The methodology sounds pretty innovative, and the methodology of the authors is well structured, based on the optimization of different operational variables. However, the fluency of the manuscript needs an overall assessment, in order to make the ratio of the experiments more easily comprehensible for the readers, and some further improvements shall also be made before the manuscript is acceptable for publication in Processes:
1) Page 2, line 48 to 50: "The complete oxidation of HMF to FDCA is not trivial; it involves three consecutive oxidation steps (Figure 1a). Because the classical synthesis of FDCA is expensive (for using metals as catalysts) and environmentally unfriendly". The authors forget to mention some bibliographycal references concerning the direct oxydation of HMF to FDCA, related to the use of metal catalysts (e.g., Fuel Processing Technology 2020, 209, 106528), as well as the other reason for which this transformation is environmentally unfriendly (e.g., high pressure, high temperatures, use of volatile and hazardous organic solvents. There are a plethora of examples in the literature, and some selected references should be provided). However, also inexpensive inorganic bases (e.g., sodium carbonate, US2015/376154, 2015, A1; lithium hydroxyde, Chem. Sci. 2018, 9, 1854 or Green Chem. 2011, 13, 824; sodium hydroxyde, Chem. Eur. J. 2013, 19,14215) or even oxygen (ChemPlusChem 2018, 83, 19) are used in aqueous media at high temperature and ambient pressure for this transformation, and this point is completely missing in the manuscript, but it deserves at least to be mentioned (with some references);
2) Page 2, line 53 to 55: "an exciting alternative is to oxidize chemoenzymatically DFF to FDCA[20,25] (Figure 1b). This type of oxidation has already been reported in batch mode with yields of less than 85%.[20,25]". This type of oxydation has been reported in batch mode also with higher yields than 85% (e.g., Green Chem. 2015, 17, 3718, 88%; in Ref. [20] the yield is even 100%). The authors should have a better look at the literature;
3) From paragraph 3.1 the manuscript is not very fluent to read: a possible improvement includes a better explanation of Figures (e.g., Figure 2 has almost no explanation), to which a reference in the main text would also be desirable (e.g., we investigated the effect of catalyst loading and H2O2 amount in the case of 10 mM substrate (Figure 2A), etc...). This should be done for Figures 2, 4 and 5 (and relative description in the manuscript);
4) Supporting information has been corrected as shown in the attached additional file.
5) English language can be checked again for minor misspells and typos

Author Response
We would sincerely like to thank the editor and reviewer for the careful revision of the paper that has permitted us to improve the overall quality of the manuscript.
Reviewer #1:
Guajardo and co-workers present an innovative biocatalytic approach, based on semicontinuous and continuous packed bed millibioreactor, for the oxidation of 2,5-furandicarboxaldehyde (DFF) to 2,5-furandicarboxylic acid. The methodology sounds pretty innovative, and the methodology of the authors is well structured, based on the optimization of different operational variables. However, the fluency of the manuscript needs an overall assessment, in order to make the ratio of the experiments more easily comprehensible for the readers, and some further improvements shall also be made before the manuscript is acceptable for publication in Processes:
1) Page 2, line 48 to 50: "The complete oxidation of HMF to FDCA is not trivial; it involves three consecutive oxidation steps (Figure 1a). Because the classical synthesis of FDCA is expensive (for using metals as catalysts) and environmentally unfriendly". The authors forget to mention some bibliographycal references concerning the direct oxydation of HMF to FDCA, related to the use of metal catalysts (e.g., Fuel Processing Technology 2020, 209, 106528), as well as the other reason for which this transformation is environmentally unfriendly (e.g., high pressure, high temperatures, use of volatile and hazardous organic solvents. There are a plethora of examples in the literature, and some selected references should be provided). However, also inexpensive inorganic bases (e.g., sodium carbonate, US2015/376154, 2015, A1; lithium hydroxyde, Chem. Sci. 2018, 9, 1854 or Green Chem. 2011, 13, 824; sodium hydroxyde, Chem. Eur. J. 2013, 19,14215) or even oxygen (ChemPlusChem 2018, 83, 19) are used in aqueous media at high temperature and ambient pressure for this transformation, and this point is completely missing in the manuscript, but it deserves at least to be mentioned (with some references);
R/ All suggested references were included on page 4 between lines 50-54.
2) Page 2, line 53 to 55: "an exciting alternative is to oxidize chemoenzymatically DFF to FDCA[20,25] (Figure 1b). This type of oxidation has already been reported in batch mode with yields of less than 85%. [20,25]". This type of oxydation has been reported in batch mode also with higher yields than 85% (e.g., Green Chem. 2015, 17, 3718, 88%; in Ref. [20] the yield is even 100%). The authors should have a better look at the literature;
R/ The literature was reviewed, that sentence was modified, and the references were incorporated in lines 58-59.
3) From paragraph 3.1 the manuscript is not very fluent to read: a possible improvement includes a better explanation of Figures (e.g., Figure 2 has almost no explanation), to which a reference in the main text would also be desirable (e.g., we investigated the effect of catalyst loading and H2O2 amount in the case of 10 mM substrate (Figure 2A), etc...). This should be done for Figures 2, 4 and 5 (and relative description in the manuscript)
R/ The requested information was incorporated on pages 5 (lines 133-139, 145- 153), 7 (lines 176- 184) and 9 (lines 195- 200) of the manuscript.
4) Supporting information has been corrected, as shown in the attached additional file.
R/ The changes were incorporated in the supporting information.
5) English language can be checked again for minor misspells and typos
R/ English language was checked and corrected.

Reviewer 2 Report
In this work, the chemoenzymatic oxidation of 2, 5-furanocarboxylic acid (DFF) to 2, 5-furanocarboxylic acid (FDCA) using semi-continuous and continuous packed bed bioreactor was explored for the first time. The best yield (100%) can be obtained by using the feed-batch reactor. The system achieved better results in fed-batch bioreactor than continuous packed-bed bioreactor and batch reactor, but it needs more data support Thus, this manuscript can be published after revised. The details comments are listed below:
1. In line 187-189, page 9, the meaning is not clear.
2. In line 179, page 8, seven h should be changed to 7 hours
3. There are many cases where H2O2 has no subscript in this paper, such as line 23, page 1.
4. In this paper, you have repeatedly suggested that the deactivation of N435 and substrates is due to hydrogen peroxide, but there is no specific experimental proof.
5. The graph annotation text needs to be unified. For example, in figure 5A, page 8, the graph annotation text in is significantly larger than the others.
6. The relationship between flow rate and residence time shown in Figure 6 is of little significance because lower flow rate necessarily implies longer residence time.
7. In figure 9, why is the biocatalyst stability not determined under optimal conditions?
Author Response
In this work, the chemoenzymatic oxidation of 2, 5-furanocarboxylic acid (DFF) to 2, 5-furanocarboxylic acid (FDCA) using semi-continuous and continuous packed bed bioreactor was explored for the first time. The best yield (100%) can be obtained by using the feed-batch reactor. The system achieved better results in fed-batch bioreactor than continuous packed-bed bioreactor and batch reactor, but it needs more data support Thus, this manuscript can be published after revised. The details comments are listed below:
- In line 187-189, page 9, the meaning is not clear.
R/ As the reviewer suggested, the paragraph was corrected. Please see lines 216-2019.
- In line 179, page 8, seven h should be changed to 7 hours
R/ As the reviewer suggested, the mistake was corrected. Please see line 207.
- There are many cases where H2O2has no subscript in this paper, such as line 23, page 1.
R/ As the reviewer suggested, the entire manuscript has been revised, and this error has been corrected.
- In this paper, you have repeatedly suggested that the deactivation of N435 and substrates is due to hydrogen peroxide, but there is no specific experimental proof.
R/ The inactivation of the biocatalyst occurs due to the high concentrations of H2O2 and by the product of the perhydrolysis reaction catalyzed by the biocatalyst Novozyme 435. This activity loss was verified in this work by studying the operational stability of the biocatalyst. In experiments, the biocatalyst was recycled several times, and the reaction yield declined from 100% in the first cycle to almost 0% at the end of the fourth reaction cycle. Please see Figure 9.
- The graph annotation text needs to be unified. For example, in figure 5A, page 8, the graph annotation text is significantly larger than the others.
R/ As the reviewer suggested, the graph annotation in figure 5A was modified like the other Figures.
- The relationship between flow rate and residence time shown in Figure 6 is of little significance because lower flow rate necessarily implies longer residence time.
R/ We agree with the reviewer, then Figure 6 was moved to supplemental information.
9) In figure 9, why is the biocatalyst stability not determined under optimal conditions?
R/ The highest yield (100%) in the fed-batch reactor was achieved at a flow rate of 3 µ/min, 100 mM DFF, 2043 µL H2O2, 1 g biocatalyst at 40 °C. According to these conditions, the operational stability tests were carried out. The only difference is that the reaction was carried out for 24 hours and not 48 hours since when performing the 6 reaction cycles the experiment would have taken a long time to complete (more than 2 weeks). There was an error in the flow rate of the reaction at 40 °C, which was corrected to 3 µL /min, and the reaction at 30 °C was performed for comparison purposes.

Reviewer 3 Report
The manuscript entitled 'Effect of operational variables on the yield of chemoenzymatic 2 oxidation of 2,5-furandicarboxaldehyde to 2,5-furandicarboxylic acid in fed-batch and continuous packed-bed millibioreactor' need further improvement especially for the data presentation. Consistency in the terms used is a must for a scientific writing. Some technical/syntax errors can be seen (e.g. the fullstops need to be after the citation). Please check carefully all the comments in the pdf file attached.

Author Response
The manuscript entitled 'Effect of operational variables on the yield of chemoenzymatic 2 oxidation of 2,5-furandicarboxaldehyde to 2,5-furandicarboxylic acid in fed-batch and continuous packed-bed millibioreactor' need further improvement especially for the data presentation. Consistency in the terms used is a must for a scientific writing. Some technical/syntax errors can be seen (e.g. the fullstops need to be after the citation). Please check carefully all the comments in the pdf file attached.
R/ As suggested by the reviewer, all points noted in the reviewer's attachment have been fixed. All changes are highlighted in yellow.
- The word "semicontinuous bioreactor" was standardized by "fed-batch bioreactor " throughout the manuscript.
- All terms were standardized throughout the manuscript.
- The word H2O2 was changed to H2O2 throughout the manuscript.
- All full stops were changed after the citations.
- In Table 1, the activity of the biocatalyst was incorporated.
- Errors in the equations were corrected.
- Figures 2, 3, and 8 were corrected.

Round 2
Reviewer 1 Report
The authors have modified the manuscript accordingly with the indications given by this reviewer in the previous evaluation step
Reviewer 3 Report
The revised version of the manuscript entitled 'Effect of operational variables on the yield of chemoenzymatic oxidation of 2,5-furandicarboxaldehyde to 2,5-furandicarboxylic acid in fed-batch and continuous packed-bed millibioreactor' has been improved by the authors. The authors have answered and clarified most of the comments raised by the reviewer previously.